# Exploring service users experiences of remotely delivered CBT interventions in primary care during COVID-19: An interpretative phenomenological analysis

Emilia Finazzi[1,2¤]*, Eilidh MacLeod[3], Angus MacBeth[1]

1 School of Health and Social Sciences, University of Edinburgh, Edinburgh, United Kingdom, 2 Department of Clinical and Counselling Psychology, NHS Grampian, Aberdeen, United Kingdom, 3 Primary Care Therapies Service, Aberdeen City Health and Social Care Partnership, NHS Grampian, Aberdeen, United Kingdom

¤ Current address: Department of Psychological Therapies, NHS West Lothian, St John's Hospital, Livingston, Scotland, United Kingdom
* emilia.finazzi@nhslothian.scot.nhs.uk

**Data Availability Statement:** All relevant data are within the manuscript. Full transcripts are available on request from the main author (EF).

**Funding:** The authors received no specific funding for this work.

## Abstract

Primary Care Mental Health Services (PMHCS) aim to provide accessible and effective psychological interventions. However, there is a scarcity of qualitative research focused on patients' experiences. Service users' experience can inform development of accessible, high-quality mental health services. Nine semi-structured interviews were analysed from Primary Care Mental Health users in Northern Scotland using Interpretative Phenomenological Analysis (IPA). Four superordinate themes were generated: Orientating to treatment, Intervention features, Change enablers, and Impact. The results identified both facilitators and barriers associated with access and psychological change; and narratives around CBT acceptability, outcomes and remote delivery. The role of GPs emerged as a key determinant of access to PMHCS. The therapeutic relationship contributed to person-centred care provision, idiosyncratic change processes and self-empowerment. A personal commitment to engage with homework was described as a crucial change enabler. Findings are discussed in relation to existing literature, practical implications and suggestions for future research.

## 1. Introduction

Globally, at least one in three individuals suffers from mental health difficulties during their lifetime [1, 2] and this statistic may be substantially underestimated [3]. The global burden attributed to mental health issues is linked to increasing social, human and economic costs due to the lack of adequate mental health services' provision and access [4–6]. Within this socio-political context, Primary Care Mental Health Services (PMHCS) play a crucial role in improving public health by providing local accessible services and interventions that meet general population needs [7]. Technology helped to develop and offer novel delivery modes of CBT interventions for various mental health conditions allowing easier, efficient and quicker access [8, 9].

**Competing interests:** he authors have declared that no competing interests exist.

Primary care services are often the initial and main provider of psychological interventions [10, 11] for mental health conditions, especially common mental health problems, such as anxiety and depression, which are prevalent in the UK and associated with substantial economic and health care impacts [12]. Blane et al. [13] provided a progress report on the commitment across all four UK nations to increase access to psychological services. In England, the 'Improving Access to Psychological Therapies (IAPT)' programme roll out started in 2008 [14, 15] allowing service users to self-refer to Primary Mental Health Care services. On the contrary, in Scotland, existing services have been redesigned following guidelines and performance targets [16] with access via GP referral. Nationwide primary care services adopted a stepped-care approach [17] where service users are offered the lowest intensity interventions first and, if needed, they are 'stepped up' to more resource intensive treatments. In line with this approach, the Matrix guidelines [18] inform both organisation and delivery of mental health services, and recommend a range of empirically validated interventions depending on the severity of the condition. For example, for mild to moderate common mental health conditions, short-term low-intensity interventions are recommended including guided self-help, one-to-one talking therapies and computerized CBT [19].

Meta-analyses of psychological therapies for anxiety and depression in primary care [e.g. 20] and other quantitative research conducted in IAPT services [21, 22] provide useful information about treatment effectiveness. For example, in the first year of the IAPT programme Gyani et al. [22] found that 40.3% of service users achieved 'reliable recovery' and recovery rates ranged between 23.9–56.5%. However, these findings do not inform us about service users' perspectives and experiences and do not provide an exhaustive multidimensional understanding of various aspects in relation to treatment and care [23]. Additionally, considering recovery in terms of 'symptom reduction' fails to include more patient-centred notions of recovery [24]. Conversely, a person-centred approach aims at deepening our understanding about what works and why, enhancing the quality and responsiveness of care provided to meet individuals' needs, with the aspiration to put the individual at the centre of decision making, through a more integrated healthcare system delivery [16, 23]. Consequently, engagement and participation of people with lived experience is deemed essential to make implementation research more relevant to individual needs [25–27], catalyse systems transformation [28] and promote inclusive and ethical practices, policies and actions [29, 30]. Therefore, exploring service users' experiences is an ethical requirement and complements quantitative findings [31] to enhance care [23, 32, 33], redesign mental health services [34], develop guidelines [23], tackle stigma [35] and improve professional training [36]. Although the importance and relevance of qualitative research exploring patients experience are increasingly recognised across mental health settings [37], qualitative research focused on service users' experiences of receiving psychological interventions in PMHCS in Scotland is limited. Indeed, only two doctoral dissertations and one mixed-method study explored service users' experience in Scottish PMHCS. Firstly, Campbell [38] investigated experiences of primary care services in service users with a diagnosis of Borderline Personality Disorder; Secondly, Hanna [39] explored older people's decision-making processes and experiences of Computerised CBT in PMHCS. Finally, Finucane and Mercer's [40] mixed-method study examined acceptability and effectiveness of Mindfulness-based cognitive therapy for recurrent depression in primary care.

On the contrary, more qualitative research conducted in England and other countries has been published [41]. A series of qualitative studies from IAPT services have explored service users' experience of low-intensity interventions for common mental health disorders [42, 43], interventions adapted to perinatal [44, 45], veterans [46], and refugees and asylum seeker populations [47]. Other studies have explored service users experience across formats such as Computerized CBT [48], online CBT [49] and group interventions [24]. LGBQ+ and Turkish-

Speaking service users' views and experiences were also investigated in IAPT services [50, 51]. Three unpublished theses examining service users' perspectives of primary care in England are also available [52–54]. Whilst Khan et al. [55] meta-synthesis focused solely on depression management in primary care services other qualitative research explored practitioners' perspectives [24, 56, 57]. The aforementioned qualitative meta-synthesis of service users' experiences of psychological treatment in primary care services [41] identified a range of factors related to structural and processual features of psychological interventions impact on treatment access, acceptability and outcome. However, none of these studies provide specific insights into service users' experience of individual psychological interventions for common mental health disorders in Scotland where PMHCS organisation differs from an IAPT model and when COVID-19 pandemic changed the delivery format of psychological interventions worldwide [58]. The aim of the present study is to fill this gap in the literature answering the following research question: how do service users make sense of remotely delivered CBT interventions delivered in a Scottish PMHCS during COVID-19 pandemic?

In particular, this study involves interviewing service users about their experiences of accessing, receiving and completing remote one-to-one psychological treatment provided by a Primary Care Psychological Therapies service in North East Scotland in the time of COVID-19.

## 2. Method

### 2.1 IPA approach

IPA is a qualitative methodology as well as an analysis method which draws on phenomenology, hermeneutic and idiography. Philosophers such as Heidegger, Husserl and Merleau-Ponty contributed to the theoretical underpinnings of this approach [59]. IPA aims to provide deeper insight into the meaning of participants lived experience, exploring idiosyncratic processes and converging experiences within a specific context [60]. IPA also acknowledges the role of researchers' reflexive interpretation of participants' subjective perceptions and unique lived experiences. IPA, as a qualitative research approach, has been used in different psychological studies to gain a deeper understanding of service users' experiences of treatment [e.g. 41, 42]. This study applied IPA throughout the entire research process to explore how patients experienced accessing and engaging in psychological therapies in a Scottish primary care service during the pandemic.

### 2.2 Setting, eligibility criteria, recruitment and participants

In accordance with national guidelines and policies [16, 18] within NHS Grampian, the Primary Care Psychological Therapies provision in Aberdeen City consists of seven Psychological Therapists who provide brief evidence based therapy to adults presenting with mild to moderate common mental health, and one Principal Counselling Psychologist and one Principal Clinical Psychologists who offer psychological interventions to individuals presenting with higher level of risk and/or more complex mental health condition (Tier 3). The Primary Care Psychological Therapies service, in which this study took place, receives approximately 3500 referrals each year from 30 GP practices across Aberdeen City [61].

All participants (N = 9) were referred to the NHS Grampian Primary Care Psychological Therapy Service by their GP due to mental health issues ranging from mild to moderate in severity and met the following eligibility criteria: individuals aged 18 and over, had completed treatment over the past 12 months and had capacity to provide informed consent. Individuals in acute crisis or experiencing significant suicidal thoughts were excluded. Clinicians working at NHS Grampian Primary Care Psychological Therapy Service identified eligible participants

**Table 1. Participants' information.**

| Participant | Gender | Age | Ethnicity | N of session attended | Primary problem when treatment started | Previous psychological treatment | Current prescribed medications for psychological problems |
|---|---|---|---|---|---|---|---|
| Jack | M | 22 | Other White | 8 | OCD | No | No |
| Fran | F | 39 | Mixed | 10 | Anxiety | No | No |
| Sophie | F | 22 | Other White | 12 | Anxiety and depression | Yes | Yes |
| Lisa | F | 44 | White Scottish | 10 | Anxiety | Yes | Yes |
| Beth | F | 54 | White Scottish | 12 | PTSD—type 1 | No | Yes |
| Kate | F | 34 | White Scottish | 12 | Anxiety | No | No |
| Nick | M | 31 | White British | 12 | Depression | Yes | Yes |
| Polly | F | 30 | White British | 8 | Anxiety | No | Yes |
| Adam | M | 36 | White Irish | 12 | Depression | Yes | No |

and informed service users who met the inclusion criteria about the opportunity to take part in the research at half-way and at the end of their agreed treatment. A leaflet with lead researcher's contact details was provided as this allowed service users who expressed interest in taking part in the study to contact directly the researcher and ask questions. The researcher (EF) sent potential participants the Participant information sheet to read and time was given to read and decide whether to take part in the research. Service users who decided to take part, were reminded that they could withdraw from the study at any time and an interview was arranged. This study' sampling was purposive and fulfilled IPA's requirement for homogeneity [62]. In qualitative research, sample size depends on a variety of factors such as richness and quality of data, the constraints under which researchers are working, study design, topic and scope [63]. This study sample size is in line with other IPA studies conducted in primary care settings [45, 46] which reached saturation. No participants withdrew their consent during recruitment or following the interview. Due to the COVID-19 pandemic restrictions all participants received psychological interventions remotely either via phone or secure NHS video call. All participants were fluent in English and able to provide informed written consent. Demographic information was gathered at the point of interview and summarised in Table 1. Six out of the nine participants were women, and eight were white. Participants' age ranged from 22 to 54 years old with an average age of 35 years old. All participants resided in Aberdeen.

## 2.3 Interview procedure

Following discussion with supervisors and review of the existing qualitative literature on this topic [41] the researcher (EF) developed the interview schedule with open-ended questions and prompts aimed to elicit rich data and limit researcher bias as recommended by IPA methodology [63]. Questions covered participants' personal experiences of accessing, receiving and completing psychological treatment. Examples of questions included "How was your experience of accessing the service?" and "How would you describe your experience of completing treatment?". Prior to the interviews, participants read the participants information sheet and completed the written consent form electronically. All interviews were conducted remotely via phone or NHS secure video calls by the researcher (EF), audio-recorded and transcribed

verbatim, with identifiable information anonymised and pseudonymized. Interviews ranged from 20 minutes to 1 hour (mean 44 minutes). To enhance quality, the research stages were supervised by a second researcher (AM) who provided feedback on qualitative interviewing after reading two initial interviews transcripts.

After obtaining ethical approval, recruitment commenced in March 2020 but paused due to Covid-19 restrictions. Minor amendments to the original interview schedule (S3 and S4 Files) were made in order to take into account the consequences of remote delivery and the unprecedented historical context of lockdown on service users experience while receiving psychological treatment. Recruitment recommenced in July 2020.

## 2.4 Data analysis

The main researcher (EF) transcribed and analysed the interviews following the six steps of IPA described by Smith and colleagues [63], and using NVIVO. (1) reading and re-reading the transcripts; (2) making initial notes; (3) developing emergent themes; (4) looking for connections across themes; (5) moving to the next transcript and (6) search for patterns across transcripts. IPA requires sustained immersion in the data which was obtained by the researcher (EF) by reading and re-reading each transcript as an individual case study and making descriptive, linguistic and interpretative comments (S1 File). After completing each individual analysis, superordinate themes and subthemes were generated and linked. Extracts from the original transcripts illustrated all subthemes. At every step of the data analysis, researchers committed to rigour and reflexivity to enhance analysis reliability and appropriateness of clustering.

## 2.5 Rigour and reflexivity

Researcher subjectivity is inherent in any research, accordingly IPA recognises the role of researchers' perceptions and interpretations to be an inseparable context framing the research process describing it as a "two-stage interpretation" or "double hermeneutic" [64]. Specifically, IPA researchers try to make sense of the participants' sense making constantly and dynamically moving between the 'part' and the 'whole' at different levels. For example, reflecting on how a participant's extract was interpreted in relation to the entire interview, and vice versa.

In this study the researcher, in line with best practice recommendations [65], maintained a commitment to transparency and reflexivity and adopted several strategies suggested by Noble and Smith [66] to enhance rigour in qualitative research. For example, the inclusion of rich verbatim participants' accounts in the findings section and the availability of interview recordings allowed the authors to review and check whether themes were grounded in the data. Furthermore, independent double coding of three transcripts was conducted by the authors and emergent themes compared and discussed. Each author reflected on their potential biases when interpreting data. To ensure trustworthiness, all researchers engaged in a shared analysis of the transcripts and in various reflexive practices. In particular, reflexivity refers to the investigation of one's own explicit and implicit assumptions, opinions and stances and how these shape and impact every stage of the research [67]. In order to be mindful and aware of subjectivity and biases and maintain transparency [68, 69] it is important to situate the primary researcher's position and experience as a trainee clinical psychologist with an interest in primary care interventions and past work experience in IAPT services. Nevertheless, her knowledge of primary care services in Scotland was limited prior to this study. In this case, researcher's remained mindful of the impact of past working experience in PMHCS on the development of the interview schedule, on the analysis and interpretation of participants' accounts as well as on representativeness of the final themes. In order to manage this, the

researcher (EF) made an active effort to remain reflective, kept a reflective diary (S2 File) to record expectations, decisions and experiences, continuously re-engaged with the transcripts and discussed bias, assumptions and interpretations in supervision [70]. Additionally, the researcher never worked at the Primary Mental Health Service where participants were recruited. This might have reduced the social-desirability bias for participants' reporting.

## 2.6 Ethical considerations

The research proposal for this study obtained the ethical approval from NHS South East Scotland B Research Ethics Committee and University of Edinburgh Health in Social Science Ethics Committee. Data were anonymised and stored in line with Edinburgh University protocols and NHS policy.

## 3. Results

Four superordinate themes related to service users' experiences in primary care were developed from the analysis: (1) Orientating to treatment, (2) Treatment features, (3) Change enablers and (4) Impact. The superordinate themes encompassed twelve subthemes described and exemplified below with participants' verbatim quotes. Fig 1 illustrates superordinate and subordinate themes. Smith and colleagues [63] suggest that measuring recurrence across cases is important especially to enhance validity in a large data set. Although there is no specific rule for 'recurrence', subordinate themes were included if they were present in at least half of the transcripts. Each participant contribution to the subordinate themes is shown in S1 Table.

## 3.1 Orientating to treatment

Experiences of accessing the service were identified across all transcripts. The majority of participants talked about the role of GPs in the referral process as well as their expectations, feelings and mindset before accessing the service. They also provided suggestions for improving access to psychological interventions.

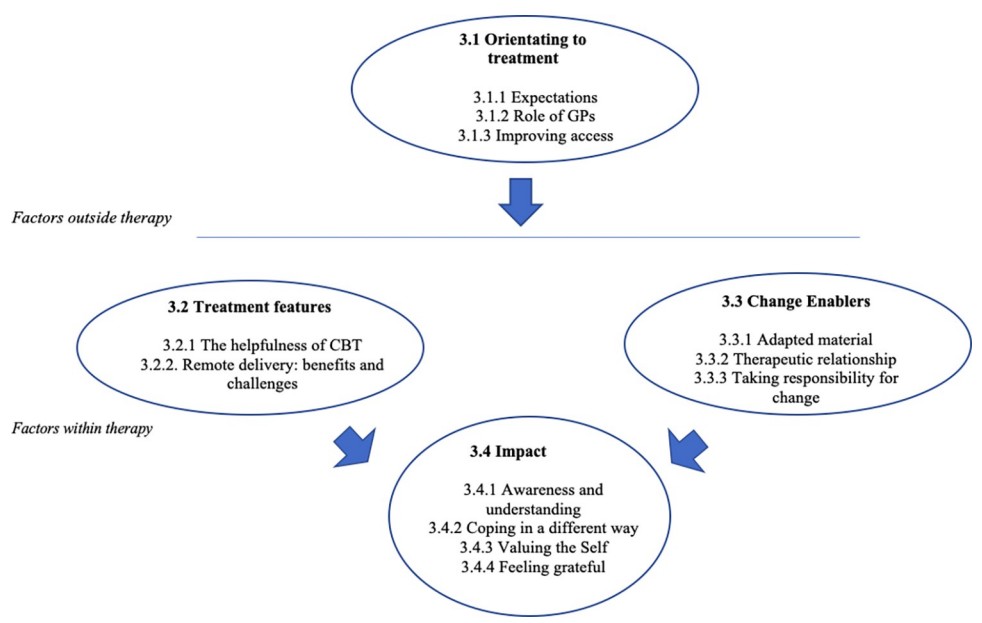

**Fig 1. Superordinate and subordinate themes.**

**3.1.1 Expectations.** Fran reported negative expectations like "this is not going to help at all", feeling hesitant or sceptical. These negative expectations were perceived as interfering with or delaying access to psychological interventions and fostering reliance on medication. Conversely, the importance of the right timing, motivation and mindset seemed to facilitate access and engagement:

*..the first time I didn't talk a lot about what's happening, it was me a lot, I did not want to get better, I just accepted everything I was just going with the flow if it makes sense and I was told to do that and I just did it (. . .)☺the second time] I said 'ok, I really need to get better' so I had that mindset while before I was just like nothing. . . if I make sense. . .The second time I just knew I had to try because I had this opportunity that other people don't and I could not give it away–Sophie*

Two participants (Adam, Jack) described realizing that the treatment offered wasn't counselling or the stereotypical 'therapy on the couch' only when starting treatment. Adam's expectations about the effectiveness of CBT were shaped by other professionals and he worried about failing as he perceived CBT to be his last resort:

*. . .there was kind of fear and there was pressure because I thought, 'if this does not work then I am fucked' excuse the language (laughs), people said this is supposed to work if it doesn't then 'what do I do?' there was probably a lot of fear but maybe if it fails I am in a lot of trouble because 'what do I do?', the experience at the start, there was quite a lot of fear there because it was so new and I was putting lot of emphasis on that working–Adam*

In Adam's case, his initial expectation of treatment was perceived as his last resort, characterised by fear of, stress and pressure to "make it work".

**3.1.2 Role of GPs.** In the context of being referred to the service and access treatment, GPs were reported as the first point of contact. The majority of participants reported a smooth and quick referral process where they felt comfortable talking to their GPs:

*She [GP] is very understanding and so. . . I had issues with my skin and with cream and things like that, and I have been to doctors and being given steroids and told to go but this one doctor she actually took the time to help and understood a bit more and I felt more comfortable to speak to her about this other issue I was having—Polly*

In the above excerpt, the fact that the GP took the time to listen seemed to be crucial for Polly to feel understood and be able to open up about her mental health difficulties. This indicated that GP's attention, time and curiosity constituted crucial ingredients for accurately assessing the patients' presentations and needs. Contrasting with the previous extract, Jack described a "weird" GP appointment, where he was reminded of the limited appointment duration and felt that he had to "prove" he qualified for therapy:

*the appointment in general was a little bit weird because I had several problems, mental health was one of them, and he actually told me "hey, listen" because I had 4–5 problems, he told me to discuss the 3 problems, the most important ones, "because we have 15 min". . . I felt I had to prove I needed therapy instead of. . .it was like "I want therapy", he [GP]: "why do you think you qualify for it?" it was very much of 'oh I need to show how vulnerable I am to get treatment' it was a little bit weird—Jack*

Although the majority of participants reported a generally positive referral experience, a few participants voiced dissatisfaction about GPs not offering psychological treatment as an option alongside medications:

> *I kind of wish I tried to do some things like therapy before this year. . .I think definitely my experience through the years is that doctors want you to be on antidepressants really quickly within the first time seeing them . . .. I guess they have 10 min with you rather than finding out what's happening they try to cover it up with medications. . . I know that antidepressants are great for some people but then what else is kind of going on, you know, that they are not really looking at, you know (. . .) I feel like GPs are quick to champion antidepressant–Kate*

The extract above echoes the challenges of brief GP appointments mentioned earlier and suggests a tendency to prescribe medications as the first line of mental health treatment, undercutting an ethos of person-centred assessment and care.

**3.1.3 Improving access.**   Seven participants provided suggestions about improving the initial stage of accessing psychological interventions as a result of reflecting on their experience. Jack and Kate's suggestions related to the role of GP as a 'gatekeeper' as something that can be changed by offering a self-referral process, improved by further mental health training for GPs, or carried out by other mental health professionals. Echoing the previous subtheme of expectations, it was suggested that providing more information at GP practices could increase knowledge of treatment options and facilitate informed choice for service users. In line with this, it was also suggested (Polly, Kate, Fran) that GPs should provide information about all options available and offer psychological treatment alongside medications sooner, especially when medication adherence is not perceived as leading to sustained and substantial improvement:

> *..when I first went to the doctor and told how I was feeling and was put on my medication, it would have been good to have being offered at that time because for years I just relied on medications and going up and down with doses, maybe feeling a bit better then go down and then feeling worse again then go up and it was not. . .dealing with the root of the problem at all, it was just kind of masking it—Polly*

Polly's extract echoed the previous theme of GPs key role in accessing psychological treatment and the dissatisfaction of considering medications as the first and only treatment for mental health issues.

Fran and Sophie also acknowledged the importance of giving choice, especially in terms of treatment modality and delivery. This was observed to allow the individuals needs and preferences to be met as well as facilitating access and engagement:

> *. . .in the future giving the patients or clients opportunities, specially people like me with ADHD or struggle with time keeping and management, maybe to give them an option of face-to-face or even phone or skype. I think that should be added..—Fran*

At the point of referral, most participants weren't told the duration of the waiting time for therapy. Four participants (Jack, Fran, Polly, Sophie) stated that the wait was much shorter than they had expected. However, two participants described a long wait of around one year, and highlighted the importance of receiving reassurance while on the waiting list to contain anxiety, feel acknowledged and maintain hope:

*Waiting a whole year and hearing nothing at all, no correspondence, you forget about it or you feel it is never going to happen, my anxiety is quite high functioning so I am still managing but I can imagine for other people it must be really hard. . ..You know what I mean? Or someone goes to the GP and asks at least a phone call straight away from them just to say "listen, we are going to do this, hang on with us, do you need anything before? Can you wait X amount of weeks for a reply?" you know what I mean to have time for you..—Kate*

In Kate's experience, a lack of contact from the service while waiting was perceived as lack of care and reassurance with a risk of escalating service users' anxiety.

Furthermore, some practical improvements were suggested such as hiring more clinicians to shorten the waiting time and allow service users to 'opt in' electronically via email as opposed to send back a physical letter in order to facilitate communication and avoid delay or unwanted circumstances.

## 3.2 Treatment features

All participants described their experience of receiving treatment remotely and how they perceived CBT modality.

**3.2.1 The helpfulness of CBT.** All participants described the CBT modality as helpful in overcoming their difficulties. In particular both Fran and Adam spoke of CBT as "what I needed".

Nick, Beth, Adam highlighted the difference between CBT and previous experience of different therapy modality suggesting that they found CBT more effective in addressing their needs. Nick, appreciated the CBT focus on behavioural change, its structure, clarity, and collaborative stance. CBT was perceived as easier to understand and to engaged with compared to other therapy approaches:

*it was good, I liked that it was all structured. . .I tried therapy before and it was open ended more discussing with the therapist I found that approach difficult, you do not know where to start while I found filing in the questionnaire that was almost a starting point for . . . you know professional help from there rather than just free style (. . .) agreeing and knowing what was happening rather than just an open ended yeah.–Nick*

Only Jack and Sophie mentioned perceived limitations of the interventions offered. Jack suggested that the number of sessions was insufficient and expressed his intention to continue CBT privately. In the extract below, Sophie described how she found the CBT tools helpful to an extent and voiced her need of talking about her past. There seems to be a contrast between what CBT offers, such as learning tools to use in future situations, and the personal need of talking and making sense of past significative events:

*I did CBT, I am not sure that it is the best thing for me, the way for me to go, although I found some stuff really helpful I do not think CBT is the thing for me in particular (. . .)I felt I needed to speak more about stuff that has happened rather than use tools to work out what is going to happen in the future (. . .) I do not think it helped me solve my problems (. . .) I know that this is the process but I also felt I did a lot of the tools they taught me they did not work for me–Sophie*

**3.2.2 Remote delivery: Benefits and challenges.** Most participants reported positive feelings and surprise at being offered commencement or continuation of treatment remotely during the COVID-19 pandemic. All participants described that the remote delivery as something

they "got used to" and already adapted to in other contexts (e.g. working from home). The adjustment to remote delivery was defined as "straightforward" (Adam), "fine" (Nick), "smooth" (Sophie) "the best way of doing it in the current circumstances" (Polly and Lisa) suggesting a quick adaptation and acceptance. Participants reported that adapting to new ways of communicating remotely did not affect significantly the overall therapeutic gains:

> *In an ideal, normal world it would have been nice to have face to face but I still certainly got a lot from over the phone–Lisa*

All participants expressed a preferred modality, with the majority of participants describing preference for face-to face. They indicated that it facilitates communication as well as building the therapeutic relationship. Particularly, being able to see the body language was mentioned as playing a crucial role in building reciprocity and human connection:

> *I just I think it is more a nurturing element to the therapy when you can actually see the person, it is a human contact whereas telephone is just a voice over the phone so if you can't physically be in the same room so this is the next best thing, you can read their body language and see how they are responding to what you are saying, you can't do that with the phone. It's more intimate, personal, so I just preferred (….) I would not prefer over the phone at all really because it does feel quite isolating–Adam*

In terms of remote delivery participants expressed a clear preference for either video or phone call. Kate opted for phone instead of video calls:

> *I probably communicated better over the phone as I am quite introverted (. . .)*

> *I preferred it (phone) (. . .)I think because one reason was that I had my sessions before work when it was face to face for first and second time, I gave all this information and felt quite overwhelmed and I had to go and work but during lockdown I wasn't working and was home and I wasn't being watched as well. It wasn't such an intimidating experience. Not that she was intimidating she was really nice yeah, it is the same thing I guess but I had more time to digest what we spoke about rather than suddenly changing how I am feeling and go to do something else, a job you know (. . .) I don't really enjoy video calls and also takes off the pressure to look good (laughs)–Kate*

Adam and Kate's modality preferences reflect two different ways of experiencing the relationship with the therapist. Adam seemed to look for a reciprocal nurturing face to face connection while Kate seemed to worry about how she was perceived and the emotional impact of the session on her.

Also Sophie found engaging in phone sessions easier. She felt more free to communicate and stay in touch with her emotions. It appeared that the lack of visual cues facilitated a disinhibiting effect:

> *I felt more free of speech because I did not have the person in front of me, I was in my room so I could react in any way I wanted, so it was easier for me to cry—Sophie*

Other participants described the practical advantage of phone delivery in terms of convenience (Lisa, Kate and Fran) of scheduling calls everywhere without taking time off or the stress associated with being on time:

*Because of my ADHD I tend to forget things I have to plan and go and then I am always late, phone sessions took all the burden away—Fran*

Nevertheless, struggles and challenges were described affecting the communication with the clinician. For example, connection problems (Adam, Nick, Lisa), not editable material electronic format (Polly), privacy issues (Polly, Kate) and the lack of body language mentioned earlier (Lisa, Fran). The extract below highlighted how the setting and the therapist's physical presence may impact on service users' concentration. It appeared that it is more difficult to maintain focus and attention when therapist and service users are not sharing the same physical space. In particular, Jack found himself more distracted at home during remote sessions:

*if I am home I would find a pen or something and play with it during the talk . . .and I think the distractions are not there when you have the face to face and I can better focus on the session—Jack*

## 3.3 Change enablers

**3.3.1 Adapted material.**   All participants reported finding the material "helpful". Some participants reported appreciating the fact that they could work on the material on their own and still use it following discharge:

*I am using some of them (worksheets) even today, yeah I feel I had to write stuff that happened, I had to write it down so I could pin point the triggers and all that kind of stuff so like it really helped me understand why it is happening and also it helped me understand when something is happening so then I can take actions so some of them really helped—Sophie*

In this case the material seemed to facilitate awareness, understanding and implementation of different responses.

Participants valued clinicians' clarity, explanations, flexibility and collaboration when agreeing homework which wasn't perceived as "excessive". Beth and Fran highlighted the importance of considering service users learning style and needs when using the material:

*Everyone reacts differently to different things and everyone has a different way of learning. I think it was very person specific, person-centred, and very specific to me (. . .)She sent me some stuff through.. I think everyone has a different learning thing, how you take things in, mine is just writing it down, I find that works for me and looking at things over and over and I remember it.—Beth*

*When I see a big thing to read I become overwhelmed and the fact that she was there and we could discuss what I understood what was there, I am a very interactive learner, so it really helped. (. . .) I need to discuss, to sink in, to hear me what I am saying and ask questions straight away from the dumbest questions to more complex ones–Fran*

In line with this, other participants highlighted the material can be a "powerful tool" when relevant and specific to the individual:

*I found the reading more beneficial than the actual exercises, the reading was excellent but there was one exercise that was excellent, that helped me cope with distress tolerance action plan I actually have that up in my bedroom wall so it just there every day, the distress tolerance action plan made a lot of sense and I am able to use it at any time I am just feeling down*

*or a bit self-destructive, that was a very powerful useful tool the other ones were a bit more writing down thoughts diary and daily planners, those things are really.. don't really work for me—Adam*

Participants reported some struggles with material related such as the burden of doing homework (Sophie, Jack) and some material not being always relevant (Adam, Sophie). Kate reported that some homework was "emotionally really hard" and "challenging" but still "helpful". Some participants valued the opportunity to access the material asynchronously at home to consolidate their learning and as a future reference to "look back at" following discharge. Sophie received signposting information relevant to her situation and interpreted that as a tangible sign of therapist genuine care.

In summary, this theme illustrated the therapist's key role in adapting and tailoring the material to each service user. It seemed that the material and homework became valuable and helpful only when the therapist made them specific and digestible to service users.

**3.3.2 Therapeutic relationship.**   All but one participant (Lisa) talked about their therapist and aspects of the working relationship. Many participants described perceived personal qualities of the therapist such as warmth, genuineness, reliability as well as their skills and professionalism.

Many participants mentioned the importance of confidentiality and being comfortable talking freely without feeling judged:

*. . .. I sometimes used very vulgar terms (laughs) and even though I would use these words the therapist never had any precaution or judgement about it I could speak freely yes also about my sexual orientation, did not even seem to bother her that was quite important.–Jack*

Participants reported not only feeling "heard" and validated but also encouraged to challenge themselves in order to overcome their difficulties. It appeared that, for Beth, receiving compassion and being challenged were both necessary for the change to happen:

*Yes very nice, very professional and like I said, very kind and caring but also pushed me a wee bit outside my boundaries, it was nice and I needed it (. . ..) I think she pushed me to go outside my comfort zone–Beth*

Other participants mentioned how they appreciated clinicians clear communication and relevant examples which helped their understanding and to normalise their feelings. Some participants emphasised the therapist's flexibility and collaborative stance which facilitated feeling included and promoted treatment acceptability:

*She included me a lot in it, asking me "does it feel right for you?"—Lisa*

*It was weekly and then we changed to fortnightly during my exams. It was helpful to have this flexibility, otherwise I could have not done it till the end.*

*(. . .) all agreeable and negotiable, everything flexible.*

*(. . .) that time was overwhelming for me but it went all smoothly because my therapist was very good as well in accommodate both of us—Fran*

Jack reflected on the contrast between the decision making process with his GP and his therapist. Specifically, the extract below seems to illustrate a contrast between a hierarchical interaction with a doctor where the patient felt like a passive recipient of treatment (e.g.

medication) and a more collaborative interaction with a therapist where the patient felt informed, involved and engaged throughout the whole process.

> *The therapist explained everything to me. She never told me what the treatment plan would be but she proposed it to me and then we would decide together (. . .) at the doctor you would just get a prescription drugs and that's it, this was more collaborative—Jack*

**3.3.3 Taking responsibility for change.** As indicated in S1 Table, five participants highlighted the aspect of "putting the work in" in order to "make it work" and for the change to happen. Taking ownership and responsibility were described as specific CBT aspects:

> *..it (CBT) was just more engaging, there was a responsibility to change behaviour whereas the person-centred it was we go in, we talk about what is troubling me and after we go "ok, see you in 2 weeks" but nothing has really changed. . .person-centred therapy was just talking there was no demand for me to do actual exercises and put it into practice, just more engaging and put more responsibility on me(. . ..) I wanted to get better but it was as much about doing the actions as saying the words–Adam*

The extract above suggests that in CBT motivation or knowledge solely appeared insufficient whilst taking action and responsibility were perceived as crucial to bring about change. Some participants highlighted the importance of pushing themselves even when it was difficult or uncomfortable such as when doing homework every week or when feeling down:

> *I definitely learnt even when I feel down I do not want to do exercise or anything like that I need to force myself to do it because I will feel a lot better afterwards so I have changed my way of thinking in that way.–Polly*

A few participants appreciated clinicians' acknowledgment of the work and helped to maintain their progress.

## 3.4 Impact

All participants described how they benefitted from treatment and the consequences they noticed in their lives and within themselves.

**3.4.1 Awareness and understanding.** In line with CBT theories, some participants highlighted how becoming more aware of their thoughts as well as identifying triggers and early signs helped them change their responses, break vicious cycles and reduce unhelpful behaviours as a result:

> *I felt more relaxed, pleasant, less anxious because I was understanding things more, the more I started to understand things the less fear was there because I don't tend to fear things I do not understand because I know what's going on. . . the drinking is not excessive anymore because I understand what it does and how difficult makes everything–Adam*

Some participants emphasised the benefit of seeing things from a different perspective, enabling them to shift the way they thought as well as implement changes:

> *well, you know you get stuck and you think the same pattern of thoughts, behaviours and beliefs and it made you think from a different perspective which makes total sense, it was*

*helpful somebody being able to point that out and put those things into practice, yeah definitely helped–Lisa*

Several participants described how relating to the examples provided helped normalise their feelings. Some participants highlighted how clinicians' validation and normalization of their feelings allowed participants to reappraise themselves and their feelings as legitimate, understandable and normal:

*She made me realised it wasn't my fault and that it was perfectly normal to feel the way I was feeling. . . That helped a lot.–Beth*

*Everyone feels sad, we all feel sad, so it is NOT abnormal, it is not an issue, and I understand that now, whereas before I thought "what's wrong with me"–Adam*

**3.4.2 Coping in a different way.** All participants reported a change in their way of coping and dealing with thoughts, emotions and feelings. The aspects of 'knowing what to do' and 'how to do it' appeared across most transcripts. It appears that CBT equipped participants with tools and strategies to implement when noticing early signs and helped them regain a sense of control over emotions and situations:

*the most important thing for me is that I know what is happening in my head and I can control it, I know the ways, what work for me and how I can control that (. . .)I know the signs when I am having a depressive phase and I know what to do–Sophie*

Polly described clearly how she started to stop overthinking and deal with her thoughts differently:

*I was stopping a lot more, stopping thinking what I was thinking about what evidence I had for it which 9 times out of 10 there is no evidence, something pops in my head and I just run with that and it is the worst case scenario and there isn't any evidence to support that thought so.. that's definitely something, even when I am thinking throughout the day thinking like a mundane task so I can focus, because I found when I am doing something it takes my mind off things when I am doing nothing I start overthinking all these things—Polly*

Of note is, coping differently was identified with a self-reported decrease in symptoms such as anxiety (Polly, Kate, Lisa, Sophie), self-harm (Sophie), substance use (Adam) as well as feeling "happier" (Adam, Kate) suggesting that the CBT tools achieved their primary goal. Additionally, some participants like Beth noticed a positive impact on their everyday life such as keeping a healthy routine or resuming daily activities (e.g. going to the shop) that they had stopped.

**3.4.3 Valuing the self.** The majority of participants reported valuing and prioritising themselves more. A change of focus and a new commitment with their Self was highlighted across transcripts:

*It helped to go back to me, you know what I mean, because at the end of the day we need to go back to us. I implemented what I learnt in CBT going back to what I knew I had to do, so for me it is fundamental I spend time outside in the nature. I need to sit and meditate every day and this I can't get rid of it. . .The focus, it really shifted, very good, a commitment with myself to do little stuff. . .of course as soon I started CBT with technique of getting my routine and*

*keep to it even if it is small, do my meditation. . . have time for myself and to be grounded and focus and everything, things can really shift. . . . . ..I try to go back to me and, if I feel all over the place, I just remind myself "did I give time to me?", "did I dedicate time to meditation, to exercise, to walk or to take care of me, or nurture me or do my things, did I organise my day what I have to do?"–Fran*

Engaging in psychological interventions also promoted a sense of control in people's lives and self-empowerment:

*it was incredible, it felt empowering, just the entire thought process, what I am worth and I remember my therapist gave me this post-it note "what do you want?" and I put it on my desk and every time I was having romance, even friendship or any activities I looked at this and I would think "do I really want this? Or am I fulfilling someone else's expectations?" so it was really interesting and I started thinking of myself as. . . that. . . I am and I look at strengths and not just my weaknesses and it was really empowering . . . I think therapy has helped me a lot as well of being a better advocate for myself.–Jack*

Finally some participants reported "feeling better in themselves" and an increased confidence which translated in practical changes in different areas of their life. For example, Polly reported doing more of what she enjoys, speaking up in groups and being proactive at work. Adam described a more positive self-image with healthier life style, habits, routine about "work, alcohol, diet, exercise".

In summary, it seemed that valuing more the Self led to practical adjustments (e.g. healthier routine), and interpersonal and intrapersonal significant changes.

**3.4.4 Feeling grateful.**   All participants reported that they found treatment helpful and that they would recommend it to others. Jack reported that he recommended it to his sister whilst Adam reported that he would recommend it to friends and colleagues "*because it really does work*" suggesting that his experience confirmed his initial expectation of CBT as an effective treatment. A few participants also reported feeling a sense of accomplishment when completing treatment.

Moreover, several participants reported positive feelings and surprise at being offered remote treatment during unprecedented times like the COVID-19 pandemic. They also reported "feeling grateful" and "lucky" for receiving psychological treatment. These feelings seemed to reflect an expectation of limited access to psychological therapy. Being offered psychological treatment was perceived as an opportunity to take rather than a regular treatment offered by the National Health Service. This aspect was highlighted by Sophie who greatly appreciated and valued the public service received as an opportunity not to take for granted:

*I think, as a person who grew up in [European country], like we don't have that at all, although I know that there are lots of problems with the NHS, but the NHS for me is something amazing so.. I felt I was really lucky. . . I should use it–Sophie*

## 4. Discussion

The current study is the only available qualitative research which offers insights into service users' experiences of accessing and receiving CBT interventions remotely in a Scottish primary care service during COVID-19 pandemic. A number of key themes emerged.

The first group of themes refer to access and orientating to treatment. Participants described the role of their expectations, ambivalence and feelings before accessing

psychological interventions. This finding is in line with other studies conducted in primary care settings [39, 52] and the previous qualitative meta-synthesis [41]. Existing research suggests that patients' expectations and attitudes are predictors of engagement [71] and outcome [72] and a small but significant meta-analytic association between pre-intervention expectations of outcomes and post-treatment outcomes [73]. This association between expectations and outcome was found positively correlated with the quality of therapeutic alliance across treatments and conditions [74, 75]. Consequently, positive expectations might promote engagement and therapeutic alliance, which may, in turn, enhance clinical outcomes. An example of this is shown in Westra et al. [76] CBT study in which early homework compliance mediated the correlation between initial expectations and early anxiety reduction. It is possible that providing information earlier could influence service users' expectations, engagement and outcomes. Interestingly, in this study only one participant mentioned stigma as a potential barrier to access psychological treatment in contrast to other studies [55, 77] and the previous qualitative meta-synthesis [41].

The role of GPs in accessing and orientating service users to treatment appeared to be crucial across transcripts. In relation to this, a survey [78] showed 40% of all GP appointments involve mental health. This study, in line with other qualitative studies [79, 80], indicated how GPs play a key role in providing a safe environment to facilitate discussions about mental health. Participants emphasised the importance of having sufficient time to talk and GPs listening to them [81].

More than half participants in this study offered suggestions for improving access. Taking into account service users' suggestions is a key element for their active participation in service improvement, research and quality care [82]. Of note, a self-referral pathway was not present in this health board area and access to psychological treatment requires GP referral. This difference might explain the relevance of GP role highlighted by service users in this study as opposed to other qualitative studies conducted in IAPT [41]. Furthermore, service users suggestions reported in this study are in line with other research and reports [83–85] indicating a urgent need of further GP training in mental health. Some participants within the study raised the proclivity of some GPs to offer medications as first line treatment although evidence-based guidelines [86] recommend otherwise for mild-to-moderate depression or anxiety treatment. More caution in prescribing antidepressant due to the severity and length of withdrawal symptoms is also emphasised in recent guidelines updates [86]. McPherson and colleagues' [87] meta-synthesis recommended the importance for patients of being informed and involved in initial discussions and decision about their treatment. Existing research showed that a significant proportion of well-informed patients preferred psychological interventions over medications [88–90]. This preference seems to be at odds with the doubled number of antidepressant prescriptions dispensed between 2008 and 2018 in the UK [91]. Furthermore, the lack of contact from the service while on the waiting list reported by a few participants echoed other research [84].

The second set of themes refer to CBT features and remote delivery. In this study participants appreciated distinctive CBT characteristics (e.g. structure, focus on behavioural change and collaborative stance) and found CBT helpful, reporting both symptom and functional improvement. These findings reflected the effectiveness of CBT as psychological treatment for depression and anxiety shown in quantitative research and meta-analyses [92, 93] in primary care settings as well [94–96]. The COVID-19 pandemic revolutionised the delivery format of psychological interventions worldwide [58] creating additional pressures for remote delivery of psychological treatments. An emerging body of literature describes challenges and opportunities of different delivery modalities [97, 98]. Participants reported adapting to the remote delivery during COVID-19 pandemic, maintaining the therapeutic gains, highlighting benefits

and reporting only a few challenges. These findings are in line with a recent literature update [99] on remotely delivered interventions for anxiety and depression which indicated good efficacy and practical benefits such as accessibility and convenience across modalities. However, evidence for video-delivered interventions was more scarce compared to other modalities (telephone and online). Telephone-administered CBT (T-CBT) has shown to be effective in meta-analysis and RCT [100–103] showing better adherence compared with face-to-face delivery but possibly at the cost of long term gain [102]. From a qualitative perspective, service user satisfaction of T-CBT was mixed [104–106]. Stubbings and colleagues' RCT [107] compared in-person CBT for mood and anxiety disorders to videoconference-based and did not find any difference. Also in terms of therapeutic alliance, no difference was found between face-to-face and both telephone [108] and videoconferencing CBT [109]. Interestingly, and consistent with emerging findings [110], three participants in this study reported symptoms worsening due to coronavirus-related stressors.

The third group of themes referred to mechanisms of change in the therapeutic process. Participants described therapist's qualities and professionalism as well as the collaborative nature of their relationship, emphasising how the therapeutic alliance allowed them to receive person centred-care. In the literature, extensive research investigated the role of clinician characteristics and therapeutic relationship in the therapeutic process [111–116]. Participants talked about the use of material and homework, which are structural aspects of CBT interventions [117] that function as important mediators of outcome [118]. Participants in this study reported benefitting from material and homework. Only a few participants did report a few homework related challenges which is unsurprising and consistent with the literature [119, 120]. Specifically, participants valued clinicians' flexibility and personalization of material and homework to meet their needs [121, 122] which, in turn, enhanced their treatment attendance and adherence as shown in other research [84, 87]. Similar to other findings [119], participants' awareness of the self as the mechanism of change and realisation that CBT interventions require a regular commitment in terms of "putting the work in" were described as crucial factors.

The last set of themes concerned the impact and consequences of psychological interventions. Participants in this study reported a variety of benefits and changes as a result of their engagement in psychological interventions. These were increased awareness and understanding, symptom reduction, coping differently and valuing themselves more. These changes are in line with a wide body of research on therapy outcomes [123–126], qualitative research [127] and the previous qualitative meta-synthesis [41]. Specifically, participants reported learning cognitive-behavioural skills which then helped them think and cope differently. Additionally, participants reported valuing themselves more and found the process of therapy as "empowering". The notion of empowerment is associated with a process of interactive teaching and experiential learning. It is focused on patients' resources and is designed to develop their awareness, knowledge and skills. Empowerment also includes an assumption that the individual will take personal responsibility and gain control over behavioural changes [128]. Some participants highlighted how psychoeducation increased their understanding of their own condition whilst clinicians' validation and examples normalised their experiences as something "perfectly normal". Normalization is a CBT technique associated with a reduction of perceived stigma and enhancement of self-esteem and coping skills [129]. Generally speaking, all participants reported a positive experience of being in treatment stating spontaneously that they would recommend it to family and friends. Service users' recommendations also echoed the Friends and family test questions used in health and mental health services [130, 131]. However, this study reached beyond patient feedback findings in offering more articulated views and suggestions for service quality improvements.

## 4.1 Strengths and limitations

This research is the first qualitative study which explored service users' therapeutic experiences of psychological interventions in a Scottish PMHCS during COVID-19 pandemic. Thanks to the IPA methodology and idiographic focus this study offered unique insights into service users' experiences highlighting common and specific factors involved in accessing and engaging in psychological treatment. A number of novel themes emerged in this study, running counter to similar research [41], such as the crucial role of GPs in accessing and orientating to treatment. In terms of quality control measures [132], a second researcher independently assessed and provided feedback on the quality of two initial interviews and all research members were involved in data analysis as well as ongoing reflection and review of transcripts. It is also important to acknowledge the researcher and interviewer role on different stages of the research process. Reflective diary extracts are accessible to provide transparency about the impact of researcher' subjectivity throughout the research (S2 File). However, this study has some limitations such as the lack of members' checking. The use of a small sample, although acceptable within IPA, recruited from one service means that findings are not representative of service users who engage in psychological interventions in primary care overall, including service users who dropped out or live in rural areas. Additionally, the sample was predominantly comprised of female, white ethnicity participants. Whilst the effect of ethnicity on service users' experience could not be explored, this sample reflects the majority of service users accessing this service in Aberdeen city where the population is 89% white [133] and 65% of patients referred to the service are female. In terms of timing interviewing considerations, not knowing how long since completing therapy each participant was interviewed represents a limitation. For example, service users who completed treatment months prior their interview might struggle retrieving memories and describe differently the treatment impact compared to those who were interviewed just after being discharged from the service.

Researcher (EF) reflected on the impact of conducting interviews remotely on the richness of the data obtained. Interestingly, Johnson et al. [134] showed that conducting qualitative interviews remotely might affect the richness of information although no significant differences were identified in interviews' length, coding and subjective interviewer ratings. It is also worth considering the volunteer [135] and social desirability bias [136] present in qualitative research which can overestimate positive features and lessen heterogeneity of the accounts. A further bias might be introduced by the fact that the sample was recruited by participants' clinicians who might hold individual biases about presentations deemed suitable to take part in the study.

## 4.2 Implications for practice and future research

The findings demonstrate the key role of GPs in accessing and orientating to psychological treatment. Therefore, it is important that GPs receive specialised training in assessing mental health needs in a non-judgemental way, being mindful of the impact of individual expectations, stigma and ambivalence towards psychological interventions. The communication about accessibility and availability of psychological interventions in primary care needs to be improved and treatment should be offered timely in accordance with the evidence based guidelines. GPs, as suggested by participants, might also provide information about psychological treatment available alongside medications. PMHCS might also provide GPs and their patients with useful information about benefits, challenges and level of commitment associated with CBT interventions. This would help people to make an informed choice and adjust inaccurate expectations that they might hold about psychological interventions. Further research is warranted to explore GPs perspectives in order to offer insights into the relational process of

help-seeking. Following service users suggestions, it may also be important to review communicating the length of the waiting list and maintaining contact while on the waiting list. Use of electronic communication (e.g. email) was also suggested instead of mail to opt-in. Further research might also explore benefits and challenges of a self-referral system in PMHCS in Scotland as it is already present in other part of the country. Given the overall positive experience of remote delivery, services should consider the option to keep offering the choice of receiving psychological treatment remotely while remaining mindful of its challenges. Findings from this research confirmed the importance of the therapeutic relationship in tailoring material and providing person-centred care which, in turn, enhanced intervention acceptability and service users' engagement. Participants' accounts also described the well-known features and outcomes of CBT interventions suggesting that treatment provided has been implemented as intended. In the early stage of treatment, allocating time to explore service users' expectations, ambivalence and views about psychological interventions and providing information to adjust expectations might support service users' access and engagement. Of note, the current study included participants who completed treatment only. Future research should explore the experiences of service users who were referred and did not access or did not complete treatment in order to gain a more holistic views of people's experiences of accessing and engaging in psychological interventions in primary care.

## 4.3 Conclusion

This qualitative study offers insight into service users' accounts of receiving psychological treatment in primary care. The themes emerged described enablers and barriers associated with access and psychological change. Some suggestions for improvement were also reported with implications for services, research and practice. For example, providing information about psychological interventions available alongside medications would help service users to set accurate expectations and make informed choice.

The findings indicated the acceptability of remote delivery and how psychological interventions facilitate awareness, understanding, psychological and behavioural change as well as empowering service users. Furthermore, therapeutic relationship was deemed crucial in tailoring material and providing successfully person-centred care. Interestingly, the aspect of taking responsibility and "putting the work in" has been acknowledged as key for effective outcomes. It is hoped that both the positive and challenging aspects of service users' lived experience presented in this study will be considered to improve access to psychological interventions, provide person-centred care and inspire future research in PMHCS.

## Supporting information

**S1 Table. Participants' contribution to themes.**
(DOCX)

**S1 File. Excerpt of transcript and IPA analysis.**
(DOCX)

**S2 File. Extract of reflective journal.**
(DOCX)

**S3 File. Original interview schedule.**
(PDF)

**S4 File. Final interview schedule.**
(PDF)

## Acknowledgments

We would like the thank the participants for their time and for sharing their experiences. The time and support of all reviewers are gratefully acknowledged.

## Author Contributions

**Conceptualization:** Emilia Finazzi.

**Data curation:** Emilia Finazzi.

**Formal analysis:** Emilia Finazzi.

**Methodology:** Angus MacBeth.

**Supervision:** Eilidh MacLeod, Angus MacBeth.

**Writing – original draft:** Emilia Finazzi.

**Writing – review & editing:** Emilia Finazzi.

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
