## [Decision Letter · Decision Letter 0]

17 Aug 2022

PONE-D-21-35164Exploring service users experiences of remotely delivered CBT interventions in primary care during COVID-19: an Interpretative Phenomenological AnalysisPLOS ONE

Dear Dr. Finazzi,

Thank you for submitting your manuscript to PLOS ONE. After careful consideration, we feel that it has merit but does not fully meet PLOS ONE’s publication criteria as it currently stands. Therefore, we invite you to submit a revised version of the manuscript that addresses the points raised during the review process.

Two expert Reviewers evaluated the manuscript. I agree with their comments, especially with the importance to use adequate terminology and more specificity when introducing the IPA method and the results according to it. I encourage Authors to submit a revised version taking into account Reviewers' comments. ==============================

We look forward to receiving your revised manuscript.

Kind regards,

Stefano Triberti, Ph.D.

Academic Editor

PLOS ONE

Journal Requirements:

2. Please include a copy of the interview guide as Supporting Information

Reviewers' comments:

Reviewer's Responses to Questions

**Comments to the Author**

1. Is the manuscript technically sound, and do the data support the conclusions?

Reviewer #1: Yes

Reviewer #2: Yes

2. Has the statistical analysis been performed appropriately and rigorously? 

Reviewer #1: N/A

Reviewer #2: Yes

3. Have the authors made all data underlying the findings in their manuscript fully available?

Reviewer #1: No

Reviewer #2: No

4. Is the manuscript presented in an intelligible fashion and written in standard English?

Reviewer #1: Yes

Reviewer #2: Yes

5. Review Comments to the Author

Reviewer #1: Introduction

General statement of prevalence and burden

Consideration to role of primary care

There then follows some interesting information. I am not sure you have considered in enough detail the past development of online interventions and I would like to see a consideration to this. What is provided talks at quite a generic level and I think being more specific is needed.

I want to know are the results you report needed is there a clear gap from past knowledge? or have they been identified previously. Looking at a brief google search there is articles that are not referenced - I also think when you consider the results the reader needs to be sure you have covered the areas considered. Eg., you talk about experience of remote delivery but a study like this considered the issues in 2009 https://onlinelibrary.wiley.com/doi/full/10.1111/j.1369-7625.2008.00531.x - I appreciate I am not an expert on CBT online delivery but it raised questions for me

Page 6

Methods

The method can begin with the methodology of IPA

I would prefer a section for eligibility criteria

When you say about all participants being referred – can you name the sampling technique used here or consider moving it

Would it be better to place demographics to results?

The first reference to IPA seems to be line 163 but I think it can come above

Can you justify how you devised the interview schedule and identify domains, then could the original interview and final one be placed in a supplementary file. Could you identify did you do a pilot or cognitive interview?

Within the analysis section can you provide an audit trail for the stages of analysis you mention in line 202 to 203

The statement around phenomenological studies is for me not needed since there are different types of phenomenology and the reader only needs to know about IPA

Consider how you achieved the double hermeneutic and hermeneutic circle

Can you reference or remove the statement that rigour refers to minimising bias?

Results

Seems ok – but I really need to come back to the introduction points around value of content compared to past studies.

Discussion

As above just considering the need and value and gap filled. I would be happy to go with a reviewer who publishes in this area more.

Reviewer #2: Abstract

Line 36; 39: While IPA is more resistant to the idea of ‘themes emerging’ given the inductive approach when collating ‘emergent’ themes, I would still be sure to use terminology here that encapsulates the researcher’s role in the development of these themes, as your interpretation was also key in these final themes. Instead of ‘emerged’ I would use ‘developed’ or ‘generated’.

Introduction

Line 88: Assumed spelling error – ‘put’ instead of ‘out’.’

Line 93 – 95: Reword this sentence, as I feel it’s a bit wordy/hard to follow.

Line 99: Maybe clarify why these papers are unpublished – how do you have access to them? Just a bit of clarity here.

Line 123 – 130: I would include an explicit reference to a research question here rather than just general aims of the research. State the question you had in mind throughout the research process/that you were attempting to address throughout the project. Link back to this question throughout your discussion.

Methods

Approach: I would generally recommend introducing IPA in an approach/research design section at the beginning of the methods given that it is a methodological approach as well as an analysis method. This would be useful as you reference IPA as a methodology throughout your sample, recruitment, participants section and within the interview procedure section, and so I feel an introduction to this as your approach is necessary at the start of your methods rather than when discussing the analysis alone.

Interview Questions: Definitely agree with the data availability in relation to the transcripts, however I would maybe recommend uploading your interview schedule/analytic trail from your analysis to a repository for some access to the processing of data.

Line 161: I would just integrate the bracketed information into the actual sentence. Reword to something like ‘Six out of the nine participants were women, and eight were White.’

Line 194: remove the full term ‘interpretive phenomenological analysis’ as you have already defined ‘IPA’ on line 163.

Line 199-200: I wouldn’t say it is necessary to state why you didn’t choose thematic analysis/grounded theory, but I would maybe argue that IPA focuses specifically on participant’s understanding/perception of their own experiences, which allows for greater depth than other analyses for your specific research question.

Line 207 – 209: Given this is related more to data collection, I would put this in the interview procedure section.

Line 210 – 2016: Linked to my previous comment on maybe introducing an approach section, I feel think would be more appropriate to either discuss there, or within the sample/participant section, as this isn’t explicitly linked to the analysis (with the exception being data saturation, but again, I think this would fit fine in your participants/data collection section).

Rigour and Reflexivity: I really like this section, and it considers the researcher’s role nicely. I would maybe insert something in here about trustworthiness when discussing rigour however as this term applies to qualitative research and IPA procedures such as yours clearly.

Line 235: To link the paragraphs together a bit more fluidly, maybe insert a sentence like: ‘To uphold rigour/ensure trustworthiness, the researchers engaged in various reflexive practices.’

Results

Line 251: See earlier comment on the term ‘emerge’.

Table 2: I’m not convinced this table adds anything to your results/findings, as you’ve already stated that all themes/sub-themes occurred in at least half the sample, and that this isn’t a requirement of IPA anyway. I do see value in it as a reference, but again, maybe something to upload to a public repository rather than being the main manuscript?

Theme labels: I would maybe re-label themes that are titled using exact quotes so that they are concise but descriptive. I find that going by Braun and Clark’s recommendation, some of these quote-named themes don’t necessarily capture the ‘essence’ of the theme itself.

Line 355: Use ‘around’ instead of ‘~’.

Line 435 – 442: I know this is a lot of information to communicate, but I would give an example/illustrative quote of some of these points.

Line 517: When presenting certain points like this, I feel there needs to be a lot more interpretation of what the participant is saying, and not just a description. What does this reflection mean? Why is it important?

Line 634: Would like to see a bit more interpretation/discussion of these terms used by the participants. To me, the idea of ‘luck’ and ‘gratefulness’ reflects a lot about the expectations of access, but again, just more interpretation of this from the researcher’s perspective would be useful.

Discussion

Findings: I think generally there needs to be a bit more interpretation of results and discussion points, especially given that it was approached from an IPA standpoint. I think certain points are quite descriptive and taken almost directly from what participants have said but leave out the important interpretation needed from the researcher. From this approach, it feels more like the discussion of a thematic analysis than an IPA in certain places (see some of my comments in the ‘results’ section).

Misc.

Check your in-text citation formatting, as they should be Vancouver formatting, and so using square brackets and commas/dashes instead of round brackets and semi-colons (https://journals.plos.org/plosone/s/submission-guidelines#loc-references).

For the sake of consistency, use either single quotation marks (‘) or double (“) when presenting in-text extracts, as you do jump between using one or the other in places (e.g., Line 445/Line 497).

6. PLOS authors have the option to publish the peer review history of their article (what does this mean?). If published, this will include your full peer review and any attached files.

Reviewer #1: No

Reviewer #2: No

---

## [Author Response · Author response to Decision Letter 0]

12 Oct 2022

Reviewer #1

Thank you for your comments. We have incorporated all your suggestions. 

- Introduction

General statement of prevalence and burden

Consideration to role of primary care

There then follows some interesting information. I am not sure you have considered in enough detail the past development of online interventions and I would like to see a consideration to this. What is provided talks at quite a generic level and I think being more specific is needed.

I want to know are the results you report needed is there a clear gap from past knowledge? or have they been identified previously. Looking at a brief google search there is articles that are not referenced - I also think when you consider the results the reader needs to be sure you have covered the areas considered. Eg., you talk about experience of remote delivery but a study like this considered the issues in 2009 https://onlinelibrary.wiley.com/doi/full/10.1111/j.1369-7625.2008.00531.x - I appreciate I am not an expert on CBT online delivery but it raised questions for me

Thank you for your comment. A paragraph about the past development of online interventions and a sentence that clarifies the gap filled have been added.

Thank you for suggesting Beattie et al. (2009) article. They used a thematic approach to explore how specifically the CBT online mode delivery impacts upon the therapeutic experience on patients with depression in a RCT in England. We appreciate the contribution offered on this topic by this paper and we added its reference in the Introduction. 

Methods

- The method can begin with the methodology of IPA. 

Thank you for your comment. The method section now begins with the methodology of IPA. 

- I would prefer a section for eligibility criteria. 

Thank you for your comment. A title section has been edited and now included “Eligibility criteria”. A paragraph in “Setting, eligibility criteria and recruitment” describes the eligibility criteria. 

- When you say about all participants being referred – can you name the sampling technique used here or consider moving it. 

Thank you for your comment. The sampling technique has been named in “Setting, eligibility criteria and recruitment”. 

- Would it be better to place demographics to results?

Thank you for your comment. Demographics and Table 1 have been moved to Results. 

- The first reference to IPA seems to be line 163 but I think it can come above

Thank you for noticing this. . There is a full descriptive section on IPA and is now at the beginning of the Method section. 

- Can you justify how you devised the interview schedule and identify domains, then could the original interview and final one be placed in a supplementary file. Could you identify did you do a pilot or cognitive interview?

Thank you for noticing this. The original interview and final one have been sent to be uploaded as supplementary files. Also, description on how the interview was devised is described at the beginning of the Interview procedure section. 

- Within the analysis section can you provide an audit trail for the stages of analysis you mention in line 202 to 203

Thank you for commenting on this. An extract of the analysis has been sent to be uploaded as a supporting file. The comprehensive and original thesis dissertation is also available at https://era.ed.ac.uk/handle/1842/38531.

- The statement around phenomenological studies is for me not needed since there are different types of phenomenology and the reader only needs to know about IPA

Thank you for noticing this. This statement around phenomenological studies has been deleted as not necessary. 

- Consider how you achieved the double hermeneutic and hermeneutic circle

Thank you for noticing this. An explanatory statement about how double hermeneutic was achieved has been added in Rigour and reflexivity section. 

- Can you reference or remove the statement that rigour refers to minimising bias?

Thank you for highlighting this. That statement has been removed.

- Results

Seems ok – but I really need to come back to the introduction points around value of content compared to past studies.

Thank you for highlighting this. A clear statement about the gap filled is available in the Discussion section. 

Discussion

- As above just considering the need and value and gap filled. I would be happy to go with a reviewer who publishes in this area more.

Thank you for noticing this. A clear statement about the gap filled is available in the Discussion section. 

Reviewer #2

Thank you for your comments. We have incorporated all your suggestions. 

Abstract

- Line 36; 39: While IPA is more resistant to the idea of ‘themes emerging’ given the inductive approach when collating ‘emergent’ themes, I would still be sure to use terminology here that encapsulates the researcher’s role in the development of these themes, as your interpretation was also key in these final themes. Instead of ‘emerged’ I would use ‘developed’ or ‘generated’.

Thank you for highlighting this. Terminology in the abstract has been edited as suggested.

Introduction

- Line 88: Assumed spelling error – ‘put’ instead of ‘out’.’

Thank you for noticing this. The spelling mistake has been corrected.

- Line 93 – 95: Reword this sentence, as I feel it’s a bit wordy/hard to follow.

Thank you for highlighting this. This sentence has been reworded. 

- Line 99: Maybe clarify why these papers are unpublished – how do you have access to them? Just a bit of clarity here.

Thank you for highlighting this. I clarified that these unpublished papers are doctoral dissertations. 

- Line 123 – 130: I would include an explicit reference to a research question here rather than just general aims of the research. State the question you had in mind throughout the research process/that you were attempting to address throughout the project. Link back to this question throughout your discussion.

Thank you for highlighting this. The research question has been stated at the end of the Introduction.

Methods

- Approach: I would generally recommend introducing IPA in an approach/research design section at the beginning of the methods given that it is a methodological approach as well as an analysis method. This would be useful as you reference IPA as a methodology throughout your sample, recruitment, participants section and within the interview procedure section, and so I feel an introduction to this as your approach is necessary at the start of your methods rather than when discussing the analysis alone.

Thank you for highlighting this. A section about IPA called ‘IPA approach’ at the beginning of Method has been added.

- Interview Questions: Definitely agree with the data availability in relation to the transcripts, however I would maybe recommend uploading your interview schedule/analytic trail from your analysis to a repository for some access to the processing of data.

Thank you for highlighting this. The interview schedule and an analytic extract and a reflective diary extract have been sent to be added as supporting files. 

- Line 161: I would just integrate the bracketed information into the actual sentence. Reword to something like ‘Six out of the nine participants were women, and eight were White.’

Thank you for highlighting this. This sentence has been reworded as suggested. 

- Line 194: remove the full term ‘interpretive phenomenological analysis’ as you have already defined ‘IPA’ on line 163.

Thank you for highlighting this. The full term has been replaced with IPA. 

- Line 199-200: I wouldn’t say it is necessary to state why you didn’t choose thematic analysis/grounded theory, but I would maybe argue that IPA focuses specifically on participant’s understanding/perception of their own experiences, which allows for greater depth than other analyses for your specific research question.

Thank you for highlighting this. The reference to other type of analysis has been removed. A paragraph explaining why IPA has been chosen to address specifically the research question is available in IPA approach section.

- Line 207 – 209: Given this is related more to data collection, I would put this in the interview procedure section.

Thank you for highlighting this. The sentence has been moved to the Interview procedure section.

- Line 210 – 2016: Linked to my previous comment on maybe introducing an approach section, I feel think would be more appropriate to either discuss there, or within the sample/participant section, as this isn’t explicitly linked to the analysis (with the exception being data saturation, but again, I think this would fit fine in your participants/data collection section).

Thank you for noticing this. This paragraph has been moved to the Setting, eligibility criteria and recruitment section. 

- Rigour and Reflexivity: I really like this section, and it considers the researcher’s role nicely. I would maybe insert something in here about trustworthiness when discussing rigour however as this term applies to qualitative research and IPA procedures such as yours clearly.

Thank you for your comment. An explanation on how trustworthiness and rigour apply to IPA has been added in Rigour and reflexivity. 

- Line 235: To link the paragraphs together a bit more fluidly, maybe insert a sentence like: ‘To uphold rigour/ensure trustworthiness, the researchers engaged in various reflexive practices.’

Thank you for highlighting this. A sentence to link the paragraphs has been added. 

Results

- Line 251: See earlier comment on the term ‘emerge’.

Thank you for your comment. This sentence has been reworded. 

- Table 2: I’m not convinced this table adds anything to your results/findings, as you’ve already stated that all themes/sub-themes occurred in at least half the sample, and that this isn’t a requirement of IPA anyway. I do see value in it as a reference, but again, maybe something to upload to a public repository rather than being the main manuscript?

Thank you for your comment. Table 2 has been removed from the main manuscript and sent to the Journal to be uploaded as a supporting Table. 

- Theme labels: I would maybe re-label themes that are titled using exact quotes so that they are concise but descriptive. I find that going by Braun and Clark’s recommendation, some of these quote-named themes don’t necessarily capture the ‘essence’ of the theme itself.

Thank you for your comment. The themes have been re- labelled without using participants’ exact quotes. 

- Line 355: Use ‘around’ instead of ‘~’.

Thank you for noticing this. The symbol ‘~’ which has been replaced with ‘around’. 

- Line 435 – 442: I know this is a lot of information to communicate, but I would give an example/illustrative quote of some of these points.

Thank you for highlighting this. Additional participants’ illustrative quotes have been added to this section.

- Line 517: When presenting certain points like this, I feel there needs to be a lot more interpretation of what the participant is saying, and not just a description. What does this reflection mean? Why is it important? 

Thank you for highlighting this. More interpretation and discussion of participants’ terms have been added in this section. 

- Line 634: Would like to see a bit more interpretation/discussion of these terms used by the participants. To me, the idea of ‘luck’ and ‘gratefulness’ reflects a lot about the expectations of access, but again, just more interpretation of this from the researcher’s perspective would be useful. 

Thank you for highlighting this. More interpretation and discussion of participants’ terms have been added in this section. 

Discussion

- Findings: I think generally there needs to be a bit more interpretation of results and discussion points, especially given that it was approached from an IPA standpoint. I think certain points are quite descriptive and taken almost directly from what participants have said but leave out the important interpretation needed from the researcher. From this approach, it feels more like the discussion of a thematic analysis than an IPA in certain places (see some of my comments in the ‘results’ section).

Thank you for your comment. More interpretation and discussion of participants’ experiences have been added. 

- For the sake of consistency, use either single quotation marks (‘) or double (“) when presenting in-text extracts, as you do jump between using one or the other in places (e.g., Line 445/Line 497).

Thank you for noticing this. Double quotation marks have been used when presenting in-text extract. Also, Vancouver reference style has been used as required.

We very much hope that we have addressed all the points raised by the reviewers and that and the edited manuscript meets the journal standards. We look forward to hearing from you.

---

## [Decision Letter · Decision Letter 1]

11 Nov 2022

PONE-D-21-35164R1Exploring service users experiences of remotely delivered CBT interventions in primary care during COVID-19: an Interpretative Phenomenological AnalysisPLOS ONE

Dear Dr. Finazzi,

Thank you for submitting your manuscript to PLOS ONE. After careful consideration, we feel that it has merit but does not fully meet PLOS ONE’s publication criteria as it currently stands. Therefore, we invite you to submit a revised version of the manuscript that addresses the points raised during the review process.

One Reviewer has suggested further modifications for the manuscript. I encourage Authors to incorporate these modifications to proceed with the editorial process. ==============================

We look forward to receiving your revised manuscript.

Kind regards,

Stefano Triberti, Ph.D.

Academic Editor

PLOS ONE

Journal Requirements:

Reviewers' comments:

Reviewer's Responses to Questions

**Comments to the Author**

1. If the authors have adequately addressed your comments raised in a previous round of review and you feel that this manuscript is now acceptable for publication, you may indicate that here to bypass the “Comments to the Author” section, enter your conflict of interest statement in the “Confidential to Editor” section, and submit your "Accept" recommendation.

Reviewer #1: All comments have been addressed

Reviewer #2: All comments have been addressed

2. Is the manuscript technically sound, and do the data support the conclusions?

Reviewer #1: Yes

Reviewer #2: Yes

3. Has the statistical analysis been performed appropriately and rigorously? 

Reviewer #1: N/A

Reviewer #2: N/A

4. Have the authors made all data underlying the findings in their manuscript fully available?

Reviewer #1: Yes

Reviewer #2: No

5. Is the manuscript presented in an intelligible fashion and written in standard English?

Reviewer #1: Yes

Reviewer #2: Yes

6. Review Comments to the Author

Reviewer #1: Thanks for addressing my concerns the details are provided are fine. Please note one reference (2009) can change. Very best wishes for your future research.

Reviewer #2: I would like to thank the authors for applying the changes recommended. I feel the following minor adjustments would assist in the final manuscript.

Methods

Thank you for addressing my comment about the Approach section. I like how this is looking, but I would still make sure you have this section focussed more on the Approach, and less on the IPA analysis. This section only needs to focus on the methodology (e.g., phenomenology e.c.t), rather than how it is actually applied. As so, you can probably remove the content from 140 – 146. I also wouldn’t put the approach section in place of the analysis of data section. I think both are still needed here, especially as the analysis of data section will allow you to go a bit more into depth about the six steps and how you applied them specifically to your study.

I would maybe also include a methods section specifically looking at your participants as this information doesn’t really count as ‘results’. I feel it would also likely fit into your setting, eligibility and recruitment section too.

There are a few nice examples of methods in these studies: https://doi.org/10.1080/0963823031000118203;
https://doi.org/10.1371/journal.pone.0265542;
https://doi.org/10.2196/mental.9934

Results

While this is more a recommendation than a must, but with the removal of the table showing participant’s contributions to themes, I would create a new table/figure which simply illustrates your theme structure. It does not need to be comprehensive as to include participant’s alignment with certain themes, but it would be useful to see your final super-ordinate and sub-ordinate themes at a glance. You could also number them in alignment with the section (see my comment on formatting). The first two DOI’s attached in my comment on methods also have tables like these which might be useful to view.

Formatting

I would say that if you’re going to number your main headings (e.g., 2. Methods) to also number your sub-headings (2.1. IPA Approach) just for the sake of consistency throughout. You could also do this with your themes (e.g., 3.2.1: The helpfulness of CBT).

7. PLOS authors have the option to publish the peer review history of their article (what does this mean?). If published, this will include your full peer review and any attached files.

Reviewer #1: **Yes: **Andrew Soundy

Reviewer #2: No

---

## [Author Response · Author response to Decision Letter 1]

28 Nov 2022

2. Reviewer #1

Thanks for addressing my concerns the details are provided are fine. Please note one reference (2009) can change. Very best wishes for your future research.

Thank you for your helpful comments and wishes. 

3. Reviewer #2

Thank you for your comments. We have incorporated all your suggestions. 

Methods

Thank you for addressing my comment about the Approach section. I like how this is looking, but I would still make sure you have this section focussed more on the Approach, and less on the IPA analysis. This section only needs to focus on the methodology (e.g., phenomenology e.c.t), rather than how it is actually applied. As so, you can probably remove the content from 140 – 146. I also wouldn’t put the approach section in place of the analysis of data section. I think both are still needed here, especially as the analysis of data section will allow you to go a bit more into depth about the six steps and how you applied them specifically to your study.

I would maybe also include a methods section specifically looking at your participants as this information doesn’t really count as ‘results’. I feel it would also likely fit into your setting, eligibility and recruitment section too.

There are a few nice examples of methods in these studies: https://doi.org/10.1080/0963823031000118203;
https://doi.org/10.1371/journal.pone.0265542;
https://doi.org/10.2196/mental.9934

Thank you for your comments and examples provided. 

A section focused on the IPA approach and a section focused on the Data analysis are now both present and distinct. Content from 140 – 146 has been moved to the Data analysis section. Information about research participants is now available in Setting, eligibility criteria, recruitment and participants.

Results

While this is more a recommendation than a must, but with the removal of the table showing participant’s contributions to themes, I would create a new table/figure which simply illustrates your theme structure. It does not need to be comprehensive as to include participant’s alignment with certain themes, but it would be useful to see your final super-ordinate and sub-ordinate themes at a glance. You could also number them in alignment with the section (see my comment on formatting). The first two DOI’s attached in my comment on methods also have tables like these which might be useful to view.

Thank you for your comments and examples provided. Figure 1 has been added to the Result section and illustrates the theme structure. 

Formatting

I would say that if you’re going to number your main headings (e.g., 2. Methods) to also number your sub-headings (2.1. IPA Approach) just for the sake of consistency throughout. You could also do this with your themes (e.g., 3.2.1: The helpfulness of CBT).

Thank you for highlighting this. The manuscript sections have been formatted as suggested.

We very much hope that we have addressed all the points raised and that and the edited manuscript meets the journal standards. We look forward to hearing from you.

---

## [Editor Report · Decision Letter 2]

5 Dec 2022

Exploring service users experiences of remotely delivered CBT interventions in primary care during COVID-19: an Interpretative Phenomenological Analysis

PONE-D-21-35164R2

Dear Dr. Finazzi,

We’re pleased to inform you that your manuscript has been judged scientifically suitable for publication and will be formally accepted for publication once it meets all outstanding technical requirements.

Kind regards,

Stefano Triberti, Ph.D.

Academic Editor

PLOS ONE
---

## [Editor Report · Acceptance letter]

28 Dec 2022

PONE-D-21-35164R2 

Exploring service users experiences of remotely delivered CBT interventions in primary care during COVID-19: an Interpretative Phenomenological Analysis 

Dear Dr. Finazzi:

I'm pleased to inform you that your manuscript has been deemed suitable for publication in PLOS ONE. Congratulations! Your manuscript is now with our production department. 

Kind regards, 

on behalf of

Dr. Stefano Triberti 

Academic Editor

PLOS ONE